# The Developing Brain: Considering the Multifactorial Effects of Obesity, Physical Activity & Mental Wellbeing in Childhood and Adolescence

**DOI:** 10.3390/children9121802

**Published:** 2022-11-24

**Authors:** Nicole E. Logan, Christie L. Ward-Ritacco

**Affiliations:** Department of Kinesiology, College of Health Sciences, University of Rhode Island, Kingston, RI 02881, USA

**Keywords:** youth, adiposity, psychopathology, anxiety, depression, physical activity, exercise, MRI

## Abstract

Obesity during childhood has been associated with many important physiological and neurological health considerations. Specifically concerning are the associations between youth obesity and declines in mental health, as shown with increasing rates of adolescent depression and anxiety worldwide. The emergence of mental health disorders commonly arises during adolescent development, and approximately half the global population satisfy the criteria for at least one psychiatric disorder in their lifetime, suggesting a need for early intervention. Adolescence is critical time whereby brain structure and functions are not only negatively associated with obesity and declines in mental health, while also coinciding with significant declines in rates of physical activity among individuals in this age group. Physical activity is thus a prime candidate to address the intersection of obesity and mental health crises occurring globally. This review addresses the important considerations between physiological health (obesity, aerobic fitness, physical activity), brain health (structure and function), and mental wellbeing symptomology. Lastly, we pose a theoretical framework which asks important questions regarding the influence of physiological health on the association between brain health and the development of depression and anxiety symptoms in adolescence. Specifically, we hypothesize that obesity is a mediating risk factor on the associations between brain health and psychopathology, whereas physical activity is a mediating protective factor. We conclude with recommendations for promoting physical activity and reducing sedentary time.

## 1. Introduction

Global rates of obesity among children have grown substantially since the mid-seventies, increasing focus on this global public health challenge. Recent estimates suggest that 7.8% and 5.6% of boys and girls aged 5 to 19 years of age, respectively, currently present with obesity [1], with 39 million children under the age of five having overweight or obesity in 2020 [2]. Increases in rates of obesity rates across this time frame has occurred alongside a small decrease in the global prevalence of moderate and severe underweight [1], highlighting that these public health issues may warrant separate and specific approaches. Specifically in the United States, recent estimates from 2017 to pre-pandemic March 2020 National Health and Nutrition Examination Survey data, show that almost 13% of children aged 2–5 years and 20.7% of children aged 6–11 years present with obesity, while 22.2% of those 12–19 years of age meet the criteria for obesity [3].

When considering the rates of obesity and related comorbidities among children it’s important to acknowledge that obesity development is multifactorial and as such may need to be examined and addressed at the individual, family, and societal levels. Recent data from samples of children in the U.S. show that race and ethnicity may play a role, as rates of obesity are highest among Hispanic youth (26.2%), followed by non-Hispanic black (24.8%), non-Hispanic white (16.6%) and non-Hispanic Asian (9.0%) children and adolescents. Sex differences also exist as boys have a higher overall and age specific prevalence of obesity compared to girls, with approximately 20.9% of boys and 18.5% of girls having obesity across all age groups [3]. Additionally, race and sex may affect some groups differently than others, as the highest prevalence of obesity is found in Hispanic boys (29.3%) and non-Hispanic black girls (30.8%). Socioeconomic status should also be considered when discussing obesity, as family income increases the rates of obesity decrease as 25.8% of children with family income 130% or less of the Federal Poverty Level (FPL) present with body mass index (BMI) values more than two standard deviations above the median, compared to 11.5% of boys and girls with a family income of more than 350% FPL. Genetic aspects of childhood obesity are also important to consider. Recent research suggests there are several genes associated with the complex aspects of obesity, including appetite behaviors, control of food intake, energy balance, insulin signaling, lipid and glucose metabolism, metabolic disorders, and adipocyte differentiation [4]. As data showing increasing global prevalence of obesity among children emerge, it becomes essential to consider the acute and long-term effects that childhood obesity may have on health and development during adolescence and adulthood. Obesity contributes greatly to healthcare costs [5], but more importantly, obesity is a considered a heritable neurobehavioral disorder that is highly sensitive to environmental conditions [6], and as such, has also been linked to metabolic and cardiovascular dysfunction and is a major risk factor for premature mortality from cardiovascular and metabolic diseases. Similarly, excess adiposity and its metabolic consequences are also linked to alterations in cognition [7], brain development [8] and mental wellbeing [9] throughout the lifespan. Adiposity is characterized by specific distributions of fat mass around the body, whereas obesity is a classification of body mass index (BMI). Such associations between physical health and brain health are important to consider within scientific and public health realms for the overall wellbeing and functioning of our youth.

## 2. Psychopathology in Youth

Depression and anxiety disorders in youth are becoming more common and represent a wider global health crisis of declining mental health. Approximately half the global population satisfy the criteria for at least one psychiatric disorder in their lifetime [10,11], and the emergence of psychopathology commonly arises during adolescent development [12]. By 2020, 5.6 million youth (9.2%) had been diagnosed with anxiety-related disorders, and 2.4 million youth (4.0%) had been diagnosed with depression [13] in the US. Adolescence is a sensitive period whereby the environment has particularly strong influences on brain and behavior, which influences daily functions such as emotion regulation and reactivity [14]. Combined, these environmental and behavioral influences put adolescents at greater risk for anxiety and stress-related disorders, which also influences structural and functional brain development [14]. Specifically, a recent review highlighted the importance of amygdala-based neural networks, the medial prefrontal cortex, the hippocampus, the striatum, and the hypothalamus in association with adolescent anxiety [15]. Importantly, these structures represent a model of neural networks which explain why adolescents are vulnerable to anxiety-related disorders, providing insight into potential brain-based approaches for targeted interventions [15]. Additionally, the anterior cingulate cortex and the limbic system have been identified as important structures and networks in adolescent depression psychopathology [16,17]. Notably, the cortical structures and neural networks associated with anxiety (amygdala, prefrontal cortex, hippocampus, striatum, and hypothalamus) and depression (anterior cingulate cortex and limbic system) are also central in the development of emotion regulation and cognitive control during adolescence [14,18] and are linked to many other daily psychological processes and abilities, including cognitive reserve, fear condition, stress regulation, motivation, and anticipatory control. Importantly, modifiable physiological lifestyle factors, such as physical activity and obesity, can also influence cognitive control, emotion regulation, and our daily psychological processes and abilities [19], which in turn, makes physical activity and obesity a prime mechanistic candidate for addressing symptoms of anxiety and depression. As such, early investigations into the adolescent brain that identify the neural correlates of psychopathology are crucial for understanding adolescent mental health, specifically in the development of adolescent anxiety and depression, and how modifiable lifestyle factors can influence this trajectory.

## 3. Considering the Complex Relationship between Physical Health and Brain Health

In parallel with rates of psychopathology increasing, rates of childhood obesity and physical inactivity have also been increasing. In 2016, four in five adolescents (81%) worldwide were considered to be living an insufficiently physically active lifestyle [20], and over 340 million children and adolescents were considered overweight or obese [2]. Previous research strongly suggests that preadolescence and adolescence is a particularly critical time whereby cognitive and brain functions are negatively associated with obesity and physical inactivity patterns [21,22,23,24,25,26,27,28,29,30]. As such, it is important to identify the extent of the contributing physiological factors to the differential patterns of brain function, brain structure, and psychopathology in youth.

### 3.1. Obesity and Adolescent Mental Wellbeing

Previous research suggests carrying excess adiposity, such as with obesity, is associated with negative mental health outcomes in adolescence. Specifically, Erermis et al. [31] identified differences between obesity severity and psychopathology patterns in early adolescence. Participants were categorized into three groups: (1) a clinical study group of 30 adolescents with serve obesity, (2) a non-clinical obese group of 30 adolescents with obesity, and (3) a control group of 30 adolescents with normal weight. Results indicated that at least half of the adolescent group with clinical obesity also had a DSM-IV diagnosis of major depressive disorder (n =16/30 participants). This group also tended to experience greater scores of anxiety-depression, social problems, social withdrawal, and adverse behavior, compared to the group with non-clinical obesity (n = 30), and the control group with normal weight (n = 30) [31], suggesting that obesity severity and psychopathology are associated during adolescence.

The association between depression and obesity might also be sensitive to female-identifying adolescents, as evidenced by prospective analyses of a community-based cohort with two decades worth of follow-up data [32]. Specifically, Anderson et al. [32] identified that adolescent obesity among females was associated with an increased risk for major depressive disorder (MDD) and anxiety disorders, with an estimated four times the risk for subsequent development of MDD or anxiety disorders compared to adolescent females with normal weight. Additionally, they reported that adolescent obesity in males is unlikely to be associated with an increased risk for MDD or anxiety disorders [32].

Furthermore, obesity was identified as a risk factor for both anxiety and depression after adjusting for demographic covariates and a family history of anxiety and depression in the Swedish Childhood Obesity Treatment Register (ages 6–17) [33]. Notably, Lindberg et al. [33] and colleagues also identified that females in the obesity cohort (46.9% of 12,507 participants with obesity) had a 43% higher risk of anxiety and depression compared to females in the general population (47.1% of 60,063 participants without obesity), a trend that was also seen in males with obesity, thus concluding that obesity is associated with risk of both anxiety and depression in children and adolescents. Similarly, a meta-analytic investigation into the prevalence of obesity in Chinese children and adolescents identified that anxiety and depression symptoms were higher in youth with higher BMI suggestive of overweight and obesity, compared to youth without overweight or obesity [34].

Lastly, recent research suggests that children with higher adiposity show associations with higher levels of trait anxiety, depression and disordered eating scores, meanwhile children with greater aerobic fitness showed associations with lower depression scores [35]. In this sample of preadolescent children, trait anxiety also mediated the relationships between higher adiposity and poorer P3 event-related potential (ERP) amplitude on an inhibitory control task, suggesting that higher adiposity and higher trait anxiety was associated with poorer brain function [35]. In sum, research suggests a higher ratio of psychopathology patterns are prevalent in youth with obesity, posing an at-risk population for adverse mental wellbeing during development.

### 3.2. Obesity and Brain Health

The development of optimal brain health during adolescence is critically linked to fostering higher-order cognitive processes, and these years are a time whereby extensive changes occur in brain structure, function and connectivity [36]. Likewise, the prevalence of obesity from childhood to adolescence is increasingly rapidly [37] and relatedly, previous research examining the association between BMI and brain morphology suggests an inverse relationship during adolescent years. Specifically, decreased volume of frontal and limbic cerebral grey matter regions are associated obesity [38]; and increased BMI is associated with a reduction in mean cortical thickness and specific bilateral cortical thickness in regions of the prefrontal cortex [39]. Additionally, recent evidence from Steegers et al. [40] reinforces the importance of maintaining a normal weight BMI through childhood, as they found evidence for an inverted-U shaped curve between BMI and brain morphology, whereby reduced gyrification (process whereby folding patterns of sulci and gyri develop on the surface of the brain) in both children with low and high BMI was evident in their sample of 3160 participants (50.3% female), in a prospective birth cohort from the general Dutch population [40]. Further, adulthood has also shown inverse associations between obesity and brain structure, with evidence of lower temporo-frontal cortical thickness in middle-aged adults with obesity [41], and reduced regional thickness, subcortical volume in young adults with obesity, compared to controls [42]. However, the association between brain morphology and aging throughout the course of the lifespan is naturally non-linear and becomes even more complex when considering obesity [43]. Notably, recent investigations into the associations between age, obesity and cortical thickness in adolescence and adulthood populations observed differential results [44]. Notably, Westwater et al. [44] and colleagues demonstrated that greater standardized BMI for age scores were associated with greater cortical thickness in adolescence, but that greater BMI for age scores were associated with reduced thickness in adulthood, such that the expected negative association between age and cortical thickness was attenuated by greater adiposity in adolescents, but augmented by elevated BMI in adults [44]. As such, further investigation into the physical and developmental correlates of brain morphology is necessary to guide neurobehavioral causes and consequences of obesity.

### 3.3. Obesity, Brain Health and Mental Wellbeing

Alongside the complex obesity-related correlates of brain health, obesity is also important in the context of psychopathology symptomology. Evidence suggests that the associations between obesity, brain health, and mental wellbeing are tightly linked, postulating that obesity alters the identifiable behavioral correlates of brain structure and function–cognition and mood [9]. Specifically, neuroimaging studies reveal differences in brain structure and psychopathology symptoms for participants with and without obesity. Chen et al. [9] observed that diffusion indices of the posterior limb of the internal capsule, corona radiate, superior longitudinal fasciculus, and lentiform nucleus, as assessed with diffusion tensor imaging (DTI), were lower in adults with obesity compared to controls with normal weight. Furthermore, these participants with obesity were also more likely to have feelings of anxiety and depression on the Hospital Anxiety and Depression Scale (HADS) [9]. In adolescent populations, BMI was negatively correlated with gray matter volume and white matter tracts [45]. Specifically, grey matter volume in the bilateral caudate, medial prefrontal cortex, anterior cingulate, bilateral frontal pole, and uncus was negatively correlated with BMI percentiles, such that greater BMI was associated with lower grey matter volume. Further, scores of positive emotions were also negatively correlated with BMI percentiles yet positively correlated with grey matter, suggesting that greater BMI was associated with lower positive emotions, and higher grey matter volume was associated with higher positive emotions [45]. Lastly, Moreno-López et al. [46] evaluated the differences between adolescents with obesity compared to controls on brain structure, reward sensitivity and impulsivity. Their results suggest that excess weight was positively associated with increased volume in the right hippocampus, and this was abnormal compared to the control group without excess weight. Further, reward sensitivity and positive urgency scores were negatively correlated with volume in the left somatosensory cortex in adolescents with normal weight, but not in adolescents with overweight or obesity [46]. As reward sensitivity is readily studied in association with obesity and eating behaviors [47], these findings have great implication for the neural basis of obesity and wellbeing-associated outcomes.

Adverse mental health symptoms commonly present during adolescence, a unique time whereby the promotion of physical health to prevent obesity, and the maturation of brain health to guide cognitive processes, is critically important for optimal development. Additionally, adolescence is also a critical time when physical activity levels decline dramatically, particularly among females, with 81% of all 11–17-year-olds self-reporting physical inactivity, with greater inactivity in adolescent females (84.7%) compared to adolescent males (77.6%) [48]. However, a notable component of mental health research is the association with early life experiences and socio-economic status (SES) [49]. Rates of childhood obesity are greater among children with a low SES upbringing [50,51]; children from low SES backgrounds also demonstrate attenuated executive functions [52,53] and cortical morphology [54,55]; and children and adolescents with low SES are at higher risk of mental health problems [56]. When assessing these components together, data from the Adolescent Brain Cognitive Development (ABCD) study indicate significant negative association between SES and BMI, as well as significant positive associations between SES and both neurocognition and cortical volume children [49]. Additionally, results indicated that BMI was negatively correlated with physical activity, and as such, suggested that future studies should identify the influence of fitness and BM on neurodevelopmental outcomes. As such, there is a need for further research on the interaction of early life experiences, physical health, brain health, and psychopathology symptom progression in adolescents.

### 3.4. Links between Obesity, Physical Activity, Brain Health and Adolescent Mental Wellbeing

Physical activity and improvements in cardiorespiratory fitness are accessible and ideal mechanisms to decrease rates of obesity in youth. Subsequently, an abundance of research suggests that cardiorespiratory fitness and physical activity are associated with positive mental health outcomes in youth [57,58]. Specifically, meta-analytic evidence from Smith et al. [59] identified strong associations between muscular fitness and self-esteem, including elements of self-esteem such as the physical self-concept, perceived physical appearance and perceived sports competence. Self-esteem is an important component of overall mental wellbeing, and is a strong predictor of anxiety and depression symptomology [60,61], thus is an important framework to consider in the development of psychopathology. Further, systematic review data suggest that higher fitness and physical activity levels are associated with better cognitive function, alongside partial support for a causal association with depression [57]. Notably, the systematic review did not show evidence for a causal association between physical activity and self-esteem in youth, suggesting the field is complex in nature. One such complexity to consider is the influence of both social and physical self-perceptions, for example, the social and physical influence of body image and obesity on the relationship between physical activity and self-esteem.

In attempt to identify the potential mechanisms behind the fitness-anxiety relationship, Williams et al. [62] proposed a model whereby perceptions of anxiety mediated the association between cardiorespiratory fitness and task-associated anxiety levels during adolescence. They suggested that higher levels of cardiorespiratory fitness were positively associated with more positive perceptions of anxiety symptoms and lower levels of state anxiety in a cohort of 185 high school students (81% female). Notably, Williams et al. [62] used an estimate rather than maximal testing to assess cardiorespiratory fitness. Shomaker et al. [63] evaluated the relationship between depressive symptoms and objectively measured cardiorespiratory fitness among adolescents with severe obesity, in attempt to characterize the influence of actual energy expenditure. In their sample of 103 adolescents, demographic factors such as age, sex, and race, as well as lean mass, were accounted for and results suggested that those with elevated depressive symptoms (16%) demonstrated poorer cardiorespiratory fitness compared to participants without elevated depressive symptoms. Accounting for lean mass in this sample of adolescents is an important strength of the study, as the outcome of relative maximal oxygen consumption (VO_2_max; mL/kg·min) obtained by maximal cardiorespiratory max fitness testing is influenced by total body weight, which has been shown to be the primary contributor to aerobic capacity in children of varying body sizes [64]. Shomaker et al. [63] conclude that future experimental studies should investigate the mediating influence of cardiorespiratory fitness on the relationship between adolescent depressive symptoms’ effect on obesity or obesity-related health comorbidities.

Considering the influence of early life experiences, such as SES, on the development of obesity, participation in physical activity, progression of mental wellbeing, and the development of brain health is crucial. Recently, the importance of considering early life social factors in the association between physical activity and depression in children has also emerged [65]. Specifically, Conley et al. [65] demonstrated that social-based physical activities are associated with lower depression scores, and that social connections partially mediate the relationship between social-based physical activities and lower depressive symptoms, in subset of participants (n = 7355) from the ABCD study (mean age = 10.01 years, SD = 7.29 months) [65]. Recent evidence from another analysis of ABCD youth suggests that regular physical activity may have extensive positive effects on the development of the functional brain connectome, and may be critical for improving the detrimental effects of unhealthy weight on cognitive health [66].

Lastly, sedentary behavior in youth populations has also been associated with lower grey matter volume in regions (frontal, parietal, occipital, and occipito-temporal regions, and the cerebellum) that also demonstrate associations with intelligence (cerebellum) [67]. This line of results suggests that sedentary behaviors or total sedentary time may influence brain structure, and thus intelligence [67]. As sedentary time is also associated with the development of obesity [68], these findings are of critical importance to physical, neurocognitive, and mental health development in youth. Consequently, physical activity and fitness promotion can improve brain outcomes and mental health during adolescence; however, further studies are needed to confirm the effects of overall physical health (e.g., physical activity, fitness, obesity) on the brain correlates of mental health in youth [58].

These areas of public health are incredibly inter-related, as physical inactivity is a leading cause of childhood obesity, obesity has been associated with adolescent depression and anxiety, physical activity has been associated with improved mental wellbeing, and specific regions, morphology and functional networks of the brain are equally associated with physical activity, obesity, and mental wellbeing. As such, it is crucial to identify the contributing risk factors, (e.g., obesity) and protective factors (e.g., fitness, physical activity), and neural correlates (brain structure and function) of adolescent mental health. In Figure 1, we postulate a theoretical framework whereby obesity and physical activity mediate the relationship between brain structure/function and psychopathology symptomology.

## 4. Promoting Physical and Brain Health

In light of what we know about the consequences of adult obesity including increased risk for cardiovascular disease, metabolic disease and mortality [70], it’s vital that effective efforts for obesity prevention are identified. More concerningly, a pattern among children with obesity has also emerged across the lifespan, as recent meta analytic data from Simmonds and colleagues [71] demonstrate that children with obesity are likely to also have obesity in adolescence that typically continues into adulthood. More specifically, 55% of obese children will present as obese adolescents and 80% of adolescents who are obese during this developmental stage will have obesity in adulthood [71]. Similar patterns emerge across the lifespan for physical inactivity as children and adolescents who are inactive during these critical life stages periods are more likely to be inactive as adults [72].

Recent estimates from the World Health Organization (WHO) show that 80% of adolescents and 27% of adults do not meet WHO’s recommended levels of physical activity [73] and that physical activity levels decline dramatically as both females and males enter adolescence [48]. Rates of obesity and physical activity are intertwined across the lifespan, as individuals who are more physically active typically have lower levels of obesity [74]. However, it should be noted that positive improvements in health outcomes result from increasing physical activity levels and decreasing obesity, are independent of one another. Therefore, increases in physical activity and reductions in sedentary behavior, independent of weight loss, are considered beneficial for a number of health outcomes [75], and should be promoted across the lifespan, specifically in childhood due to the positive effects consistently demonstrated from increased activity levels and physical fitness on brain health and cognitive performance [76]. As such, the promotion of physical activity and the reduction of sedentary time should be prioritized early in development to optimally enhance physical health, and consequently, brain health and mental wellbeing. The ideal setting and execution for promoting physical activity among youth has not been elucidated but as education is compulsory in many countries, formal and informal educational settings may be an optimal location for physical activity promotion for children and adolescents [77]. These programs can be incorporated into already present physical education programs, through in class instruction or play/activity breaks.

Given the positive associations for brain health after acute bouts of physical activity in preadolescents [78], interventions which focus on providing acute bouts should be promoted when possible. Brief, acute bouts of exercise in education settings are strong candidates for intervention methods, as perceived time constraints and location restrictions are commonly identified barriers to exercise adherence in children and adolescents [79]. Further, the aggregation of acute bouts of exercise throughout the day helps to meet the recommendations from current physical activity guidelines for 60 min of moderate-to-vigorous physical activity per day [80].

The in-school promotion of physical activity is a unique avenue whereby children are taught healthy lifelong behaviors to implement as part of their regular daily routine, but also gives children the opportunity to take home the important message of incorporating physical activity into the day with family members [81]. Further, after-school and community-based physical activity and education programs which target socialization and enjoyment for the whole family. Further, after-school and community-based physical activity and education programs which promote physical activity and its associated outcomes can be beneficial for whole family units, as caregivers can model more optimal lifestyle behaviors [82,83]. Community-based activity and educational programs also provide additional avenues for social support and enjoyment [84], which are important psychosocial determinants of youth physical activity adherence [85,86] and further predict adult participation in physical activity [87]. Consequently, the promotion of physical activity programs in formal educational or community-based settings are prime candidates for family-wide adherence to healthier life-long behaviors.

## 5. Conclusions

Considering the concerning rates of youth obesity and suboptimal mental health worldwide, especially in high-risk populations at the individual, family, and societal levels, it is increasingly important to establish early and cost-effective intervention or treatment options, such as with physical activity, which promote lifelong adherence to healthier lifestyles. This review discusses the important influence of cortical structures and neural networks which are associated with (i) adolescent psychopathology, (ii) daily cognitive and emotion regulation processes, and (iii) attenuated cognition, brain function and structural characteristics in youth with obesity and low physical activity levels. Further, we pose a theoretical framework whereby obesity and physical activity mediate the relationship between brain structure/function and psychopathology symptomology. Lastly, we recommend the promotion of physical activity programs in formal educational or community-based settings as prime candidates for family-wide adherence to healthier life-long behaviors. As data continue to show increasing global prevalence of childhood obesity, it becomes essential to consider the acute and long-term effects which childhood obesity may have on brain health and development during adolescence and into adulthood. As such, the public health intersections between childhood obesity, physical activity, brain health and adolescent mental health is ripe for further scientific investigation.

## Figures and Tables

**Figure 1 children-09-01802-f001:**
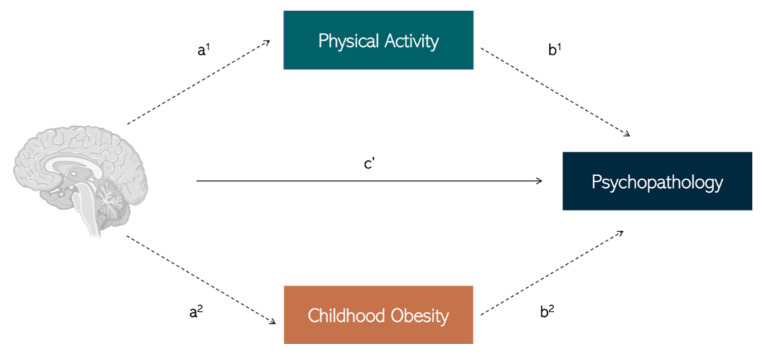
Hypothetical mediation model whereby the relationship between brain structure and function and psychopathology (c’ path) is mediated by protective factors such as physical activity (a^1^-b^1^ path), and/or risk factors such as childhood obesity (a^2^-b^2^ path). Created with BioRender.com [69], accessed on 25 October, 2022.

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
