# Peer review of "The Developing Brain: Considering the Multifactorial Effects of Obesity, Physical Activity & Mental Wellbeing in Childhood and Adolescence"

_children, 2022, doi:10.3390/children9121802_

Round 1
Reviewer 1 Report
1) In abstract, please talk more about the theoretical framework which has been addressed in this review.
2) Please use a same word. Obesity or adiposity.
Good luck
Author Response
Comments and Suggestions for Authors
Reviewer 1:
- In abstract, please talk more about the theoretical framework which has been addressed in this review.
Thank you, we have added further explanation of the theoretical framework into the abstract (lines 22-25).
- Please use a same word. Obesity or adiposity.
Thank you for this comment. Obesity refers to a classification of weight status, however, adiposity refers to the distribution or type of adiposity assessed throughout the body. Different studies reported in this review have assessed the distribution/type of adipose (adiposity), or the weight status of individuals (i.e., overweight, obesity). As such, they are not interchangeable terms, but a definition of adiposity has been added into the manuscript (lines 65-69).
Good luck
Reviewer 2 Report
Thanks for the opportunity to review this review. It was a joy to go over the manuscript. I do think it is a well-written thorough review with details. A suggestion that I would like to give is more about the genetic side of obesity.
As the text also underlines obesity/overweight problems that develop in early childhood years are generally persistent through adolescence and adulthood years and are linked to physiological and psychological problems. So, it is important to detect the predictors of obesity/BMI in early years. The review is covering most of the potential predictors. However previous research also revealed the importance of genetic factors related to early BMI/obesity. Genetic factors are not covered in this review. It can be a point to complete the puzzle.
Author Response
Reviewer 2:
- Thanks for the opportunity to review this review. It was a joy to go over the manuscript. I do think it is a well-written thorough review with details. A suggestion that I would like to give is more about the genetic side of obesity. As the text also underlines obesity/overweight problems that develop in early childhood years are generally persistent through adolescence and adulthood years and are linked to physiological and psychological problems. So, it is important to detect the predictors of obesity/BMI in early years. The review is covering most of the potential predictors. However previous research also revealed the importance of genetic factors related to early BMI/obesity. Genetic factors are not covered in this review. It can be a point to complete the puzzle.
Thank you for this important comment. We have considered a recent review on obesity genetics in childhood, and added this information intro the introduction (lines 55-58).